# Voltage-Dependent Anion Channels in Male Reproductive Cells: Players in Healthy Fertility?

**DOI:** 10.3390/biom14101290

**Published:** 2024-10-12

**Authors:** Stefano Conti Nibali, Giuseppe Battiato, Xena Giada Pappalardo, Vito De Pinto

**Affiliations:** Department of Biomedical and Biotechnological Sciences, University of Catania, Via S. Sofia 64, 95123 Catania, Italy; stefano.continibali@unict.it (S.C.N.); giuseppe.battiato@phd.unict.it (G.B.); xenagiada.pappalardo@unict.it (X.G.P.)

**Keywords:** voltage-dependent anion channel (VDAC), sperm, male fertility, mitochondria, DNA methylation

## Abstract

Male infertility affects nearly 50% of infertile couples, with various underlying causes, including endocrine disorders, testicular defects, and environmental factors. Spermatozoa rely on mitochondrial oxidative metabolism for motility and fertilization, with mitochondria playing a crucial role in sperm energy production, calcium regulation, and redox balance. Voltage-dependent anion channels (VDACs), located on the outer mitochondrial membrane, regulate energy and metabolite exchange, which are essential for sperm function. This review offers an updated analysis of VDACs in the male reproductive system, summarizing recent advances in understanding their expression patterns, molecular functions, and regulatory mechanisms. Although VDACs have been widely studied in other tissues, their specific roles in male reproductive physiology still remain underexplored. Special attention is given to the involvement of VDAC2/3 isoforms, which may influence mitochondrial function in sperm cells and could be implicated in male fertility disorders. This update provides a comprehensive framework for future research in reproductive biology, underscoring the significance of VDACs as a molecular link between mitochondrial function and male fertility.

## 1. Introduction

Infertility, defined as a couple’s inability to conceive after at least one year of unprotected intercourse, is a problem that affects about 10–15% of couples worldwide [1,2]. A male factor is involved in about 50% of couples, either isolated (30–40%) or combined with a female factor (10–20%) [3]. There are several causes of male infertility, which can be extensively classified according to their general etiology. These include endocrine disorders [4], sperm transport disorders [5], and primary testicular defects [6]. Other factors affecting the ability to conceive may include age, genetic problems, systemic diseases, medications, surgical history, and exposure to environmental toxins [7,8,9].

Mammalian spermatozoa are highly differentiated haploid cells that are unable to or have only limited capacity to synthesize proteins de novo. Spermatozoa undergo sequential maturations. Most of the components found in mature spermatozoa are primarily formed in spermatids and, therefore, developing spermatids manifest a variety of morphological and biochemical changes. Many of the organelles in spermatids are transformed into specific structures such as the acrosome and the axoneme. The fundamental proteins are integrated into these structures or associated elements. At the end of spermatogenesis, upon spermiation, most of the cytoplasm and cell organelles are discarded from the nascent spermatozoa. In their migration from the testis to the distal epididymis, spermatozoa change from immotile and immature (infertile) to motile and mature (fertile) cells [10]. This indicates that fundamental proteins are programmed to be synthesized promptly, precisely transported, and integrated into the structures and elements in developing germ cells. Such molecular events are coordinated with cell organelle behavior [11].

Spermatozoa are highly specialized cells in which the roles of mitochondria are essential for male fertility [12,13]. Indeed, even though spermatic cells can survive by glycolytic energy, they predominantly depend on oxidative metabolism for their normal physiology [14,15]. Sperm mitochondria, localized in the midpiece, are essential for germ cell function by producing energy, maintaining redox equilibrium, and regulating calcium metabolism, all necessary for sperm movement, capacitation, acrosome reactions, and sperm–oocyte fusion [12,14]. Although sperm mitochondria are not transmitted to the descendant, their abnormal function and/or genetic defects in mitochondrial DNA (mtDNA) might compromise sperm physiology and motility, causing problems with spontaneous conception [16,17,18].

Voltage-dependent anion selective channels (VDACs) are multifunctional channels present across all eukaryotes’ outer mitochondrial membrane (OMM). In mammals, highly conserved through evolution, three distinct genes encode for three VDAC isoforms: *VDAC1*, *VDAC2*, and *VDAC3* [19,20]. The isoforms are characterized by similar molecular weights (28–32 kDa) and by approximately 70% sequence similarity. Despite the high sequence homology, VDAC isoforms display different roles in physiological and pathological conditions, expression levels, and tissue specificity [21,22,23]. Even though VDACs have been extensively investigated, only a few studies focus on VDAC proteins in male germinal tissues and sperm cells [24,25,26,27,28,29], which have suggested that VDACs might play important roles in spermatogenesis, playing a critical role in male fertility. Nevertheless, the exact role and function of VDACs in mammalian spermatozoa during reproduction are yet to be fully understood.

Given the importance of male fertility, this review summarizes the literature to understand the involvement of VDAC proteins in spermatogenesis and male fertility and the association between VDAC expression variation and male infertility/poor sperm motility.

## 2. Overview of Mitochondria in Spermatozoa

Mitochondria are widely recognized as the primary drivers of cellular energy production. They are highly dynamic organelles characterized by two membranes—the outer and inner mitochondrial membranes (OMM and IMM)—which delineate the intermembrane space (IMS) and mitochondrial matrix. Mitochondria exhibit specialization in shape and structure according to cell type and energy demands [30]. Additionally, they can swiftly respond to cellular environmental stress by altering their shape and modulating activities [31]. Mitochondrial quality control mechanisms, involving continuous fusion and fission, along with the balance between mitochondrial biogenesis and mitophagy, enable the modulation of mitochondrial structure based on cellular energy requirements [32]. Indeed, mitochondria play a crucial role in cellular homeostasis by ensuring bioenergetics, biosynthesis, managing byproducts, calcium handling, and apoptosis [33,34].

Mitochondria are pivotal for male fertility, contributing to spermatogenesis and oocyte fertilization. During spermatogenesis, mitochondrial function supports germline proliferation and differentiation, with mitochondria changing in size, number, and shape in association with different stages. Notably, early-stage mitochondria are small with limited oxidative capacity, while later stages see mitochondria becoming more condensed and elongated, with enhanced OXPHOS capacity [35]. Specialized spermatozoa production involves the loss of most mitochondria, with some arranged in tail microtubules to provide energy for sperm motility [35]. Furthermore, sperm mitochondria contain specific isoforms of proteins and enzymes, such as cytochrome c [36], subunit VIb of cytochrome c oxidase [37], lactate dehydrogenase (LDH-X) [38,39], and E1-pyruvate decarboxylase [40]. These unique isoforms distinguish the functional characteristics of sperm mitochondria from those of somatic cells. LDH-X, in particular, is localized in both the mitochondrial matrix and the cytosol [41]. The interaction between the lactate carrier and the cytosolic and mitochondrial isoforms of LDH enables the efficient transport of reducing equivalents from the cytosol into the mitochondria. Therefore, the presence of specific lactate dehydrogenase isoforms makes lactate a key substrate for sperm mitochondria, alongside pyruvate [42].

The regulation of Ca2+ influx into spermatozoa is crucial for capacitation, acrosome reactions, and fertility [43]. The fallopian tube environment in the female reproductive tract serves as a major source of calcium ions, with Ca2+ influx regulated by various ion channel proteins in the plasma membrane. Additionally, mitochondria play a role in calcium homeostasis, raising questions about the involvement of sperm mitochondria in calcium signaling. Research by Bravo et al. highlighted the role of the mitochondrial calcium uniporter (MCU) in sperm physiology as blocking the carrier reduced sperm mobility and ATP levels [44]. Furthermore, calcium regulation is essential for mitochondria-dependent apoptosis; an overload of mitochondrial calcium leads to the release of caspase cofactors and ultimately apoptosis [45].

For decades, it was generally believed that the translation of nuclear-encoded proteins did not occur in mature sperm cells. However, studies have shown that mammalian spermatozoa contain nuclear-encoded mRNAs [46] and have the ability to synthesize mitochondria-encoded RNA and proteins [47]. Gur and Breitbart revealed that mitochondrial ribosomes in spermatozoa translate nuclear-encoded proteins, indicating that protein translation might be essential for sperm functions, including motility, actin polymerization, and the acrosome reaction, which are critical for fertilization [48].

## 3. Mitochondria Dynamics during Spermatogenesis

Mammalian spermatogenesis is a complex process orchestrated by spermatogonial stem cells (SSCs) and unfolds within the convoluted seminiferous tubules of the testes [49,50]. Briefly, spermatogonia, the progenitor cells, undergo mitotic divisions to expand the SSC pool [51]. During mitosis, daughter cells remain interconnected, forming syncytial cells, a phenomenon facilitated by surrounding somatic Sertoli cells that provide structural support and metabolites. Undifferentiated spermatogonia transition into type A1 differentiating spermatogonia upon receiving differentiation signals from Sertoli cells. Subsequently, A1 spermatogonia progress to type B spermatogonia, which then differentiate into meiotic spermatocytes within the adluminal compartment. During meiosis, a spermatocyte undergoes two divisions to generate four haploid round spermatids [52]. These spermatids undergo morphological changes in a process called spermiogenesis, involving nuclear condensation, acrosome formation, and tail development.

The mitochondrial network undergoes significant alterations throughout spermatogenesis. Initially, spermatogonia feature small, spherical mitochondria dispersed in the cytoplasm, categorized as the “orthodox” type with low oxidative phosphorylation (OXPHOS) activity, relying on glycolysis for energy production [35,53,54]. As spermatogonial differentiation progresses, mitochondrial dynamics shift. Mitochondria in meiotic spermatocytes aggregate around the nuage, forming intermitochondrial cement (IMC), and exhibit elongation with a condensed cristae ultrastructure, indicative of increased OXPHOS reliance [35,54]. In haploid spermatids, mitochondria display an intermediate structure between condensed and orthodox types, with a cytoplasmic distribution [35,54]. Towards the end of spermiogenesis, mitochondria elongate and finally coil within the midpiece of mature spermatozoa [54]. The role of VDAC2 during spermatogenesis was reported by Fang et al., showing that the downregulation of VDAC2 inhibits spermatogenesis via the JNK/P53 cascade [55]. Recently, Shimada et al. highlighted the involvement of armadillo repeat-containing 12 (ARMC12) as a key protein for mitochondrial sheath formation. In fact, the absence of ARMC12 leads to abnormal mitochondrial coiling, reducing sperm motility. The authors found out that AMRC12 interacts with VDAC2 and VDAC3, which act as scaffolds to link mitochondria, regulating the mitochondrial dynamics during spermatogenesis [56].

## 4. VDACs’ Role in the Mammalian Testis and the Spermatozoa

VDACs are required for the metabolic cross-talk between cytosol and mitochondria, allowing the passive diffusion through the OMM of metabolites up to ∼5 kDa, including ATP/ADP and nucleotides, NAD+/NADH, and many intermediates of the Krebs cycle (e.g., glutamate, pyruvate, succinate, malate) [57,58,59,60]. VDAC proteins also regulate the flux of small ions (Cl^−^, K^+^, and Na^+^) and participate in fatty acid transport and cholesterol distribution in mitochondrial membranes [61]. Furthermore, VDACs maintain the physiological level of cytosolic calcium and are considered the main escape route for mitochondrial superoxide anions (O^2−^) to the cytosol [62,63,64]. Beyond their metabolic functions, the peculiar position of VDACs at the interface between the cytosol and mitochondria makes them the mitochondrial docking site for several cytosolic proteins (e.g., hexokinases, glycerol kinase, glucokinase, and creatine kinase) [65,66], including molecules involved in the regulation of cell life and death. In this perspective, VDACs are crucial for maintaining mitochondrial bioenergy and communication between the organelle and the rest of the cell [19]. Even though VDACs have been extensively investigated, only a few studies focus on the localization, expression, and functions of VDAC proteins in male germinal tissues and sperm cells. In such specialized and highly developed cells, VDACs appear to exhibit new cell localization and unusual functions.

The interest in the role of VDACs in spermatozoa originates from the results obtained by knocking out each isoform in mice. VDAC1-deficient mice are viable and fertile with mitochondrial functions slightly affected, dependent on the strain and tissue examined [67,68]. VDAC2 gene deletion is lethal and only VDAC2-deficient stem cells have been obtained [69]. Recently, Chin et al. showed that viable male mice at birth exhibit small testes lacking mature spermatozoa but with expanded numbers of spermatogonia and the appearance of giant cells in the seminiferous tubules. The authors suggested that the phenotype found may be related to the altered regulation of the apoptotic process regulated by VDAC2 and its interaction with the BCL-2 protein family [70]. Moreover, Liu and co-workers analyzed the expression of VDAC isoforms between normozoospermic and infertile donors, showing a positive correlation between a high abundance of VDAC2 expression with low sperm motility and highlighting that the overexpression of VDAC2 might be related to idiopathic asthenozospermia [28].

Mice lacking VDAC3 are healthy, but males are infertile [26]. Although there are normal sperm numbers, the sperm exhibit markedly reduced motility. VDAC3-deficient males have normal numbers of sperm per epididymis in comparison to wild-type and heterozygous males and show no significant differences in the size, weight, or histologic features of testes. However, whereas in wild-type and heterozygous groups, ~70% of the sperm are categorized as motile, only 17% of VDAC3-deficient sperm are categorized as motile [26,68]. When viewed by electron microscopy, 68% of the VDAC3-deficient epididymal sperm axonemes (versus 9% of the wild-type axonemes) in a cross-section demonstrated structural aberration, most commonly the loss of one outer doublet from the normal 9+2 microtubules doublet arrangement. In the majority of VDAC3-deficient axonemes, the missing doublet corresponds to the last of the four doublets (doublet 7), reflecting a single recurring defect in the axonemal structure. Electron microscopy of spermatids in the testes revealed enlarged and abnormally shaped mitochondria along the midpiece [26]. Infertility due to sperm immotility may thus be a consequence of axonemal defects. Recently, Mise et al. confirmed that male VDAC3-deficient mice are infertile due to reduced sperm motility, potentially linked to an abnormal mitochondrial sheath in spermatozoa and increased spermatozoon bending. Furthermore, the authors identified two novel interacting partners of VDAC2/3, the Kastor and Polluks polypeptides, which may play a role in mitochondrial sheath formation. However, further studies are needed to clarify the details and underlying molecular mechanisms [71]. The information from KO mice pointed to the main role of VDAC3 in sperm fertility.

## 5. VDAC Isoform Expression and Localization in the Mammalian Testis and the Spermatozoa

Based on protein detection, specific VDAC isoforms are observed in specialized cells or compartments in the mammalian testis and spermatozoa. Mammalian studies indicate that VDAC1 is found in regions supporting gamete development, i.e., Sertoli cells [24,72]. VDAC1 is also identified in bovine sperm, although its cellular localization is not cleared [25]. Unlike isoform 1, transcriptional investigation and protein localization analysis have revealed that VDAC2 and VDAC3 are present in germ cells and testis. An mRNA analysis showed that VDAC2 is expressed in bovine testis, and high levels of VDAC2 proteins were found in late spermatocytes, spermatids, and spermatozoa [25]. Specifically, through the histochemical staining of bovine testis tissue, VDAC2 was detected in spermatocytes and spermatozoa but not in Sertoli cells [24]. Specific anti-VDAC antibodies were used for a preliminary investigation of their respective antigens in bovine sperm. Immunofluorescence microscopy clearly showed that the VDAC2 protein is present in the sperm tail. In addition, commercial antibodies against phosphorylated Tyr recognized VDAC2 spots in 2D-PAGE [73,74]. These results outlined that VDAC2 is present in cells of the spermatogenic line and that covalent modifications occur in these proteins and most likely are involved in the maturation of the spermatozoa. Liu and co-workers [75] demonstrated that VDAC2 is localized in sperm flagella and the mitochondria’s outer dense fibers (ODFs), preserving sperm tail structural integrity through microtubule-associated sperm mobility [25]. However, VDAC2 is also found in the plasma membrane or acrosomal membrane of the sperm head [27], thereby being able to participate in the acrosome reaction. Hints about the role of VDAC2 were, however, reported by Xu et al., who showed that alterations of VDAC2 expression is correlated with male infertility in patients with idiopathic asthenospermia [76].

VDAC3 is present in all testis cell types, especially in Leydig cells [77]. In mature spermatozoa, it is also abundant in the bovine and mouse spermatozoa, where VDAC3 is present in the acrosomal region, tail, and the ODFs of the flagella, which indicates that it can also be located in non-membranous components [25,27,28]. Furthermore, it has been reported that genetic variation of the VDAC3 gene is related to diminished semen quality in males with definite idiopathic infertility [78], and male mice lacking VDAC3 show markedly reduced sperm motility and are infertile [26]. Overall, this suggests that VDAC3 may be very important for male fertility.

The localization of VDAC2 and VDAC3 in the ODFs of the sperm flagellum is peculiar [25]. The sperm tail, indeed, can be subdivided into four compartments: neck, midpiece, principal piece, and end piece. In particular, the midpiece contains all the mitochondria surrounding the ODFs and the axoneme, and the principal piece contains the fibrous sheath, ODFs, and the axoneme. ODFs have a distinguishable, characteristic shape, contain a cortex surrounding a medulla, and are highly insoluble. Functionally, ODFs have been suggested to maintain the passive elastic structure and elastic recoil of the sperm flagellum and possibly also protect it from shearing forces during epididymal transport. ODF proteins were extracted and limited proteolysis and MALDI-TOF analysis identified peptides from VDAC2 and VDAC3 [25]. VDAC isoforms have also been identified in Leydig cells, a type of interstitial cell responsible for androgen biosynthesis. These isoforms interact with the Translocator Protein (TSPO) within an 800 kDa protein complex, as determined by Blue Native-PAGE (BN-PAGE). This complex also includes IMM proteins such as AAA+ ATPase ATAD3A and cytochrome P450 side-chain cleavage enzyme (CYP11A1). Cholesterol, the precursor for all steroid hormones, is proposed to be imported into mitochondria through a mechanism involving VDACs and TSPO. Consequently, VDAC isoforms appear to play a crucial role in the initiation of steroidogenesis [79,80,81]. These differences in expression and localization may indicate distinct structures and functions of VDAC isoforms in mammalian germinal tissues and spermatozoa, although the subcellular localization of VDAC isoforms in spermatozoa deserves further investigation.

## 6. VDACs and Sperm Capacitation and Acrosome Reaction

An intact acrosome is a prerequisite for normal acrosome reaction and sperm–egg fusion [82]. It is generally agreed that the acrosome reaction is a Ca^2+^-dependent mechanism, and this physiological event positively correlates with the intracellular calcium concentration [83,84]. VDACs in somatic cells contain a Ca^2+^ binding site and regulate calcium transmembrane transport, suggesting that VDACs in the sperm membrane may regulate the acrosomal integrity and acrosome reaction through mediating Ca^2+^ transmembrane flux. Specifically, VDAC2 has been discovered in the acrosomal membrane of bovine sperm heads [24], and it has been suggested as a key player in the acrosomal process. Liu and co-workers demonstrated that the co-incubation of bovine spermatozoa with anti-VDAC2 antibody causes an increased loss of acrosomal integrity and changes in the sperm head morphology, which are presumably due to the alteration of the intracellular ion concentration [75]. Furthermore, VDAC2 is also involved in another process mediated by the intracellular calcium concentration, sperm capacitation, a complex process involving a series of molecular changes required for sperm to fertilize the oocyte [85]. Ion channels play a crucial role during sperm capacitation, allowing specific molecules essential for achieving this physiologic status to be transported through the plasma and mitochondrial membranes [85,86]. VDAC2, thanks to its localization, might regulate calcium transport and distribution and play roles in sperm capacitation. Using different VDAC2 inhibitors (i.e., erastin, olesoxime, and 4,4′-diisothiocyanostilbene-2,2′-disulfonic acid) has demonstrated the importance of this isoform in the processes required to fertilize the oocyte, confirming that VDAC2 is highly implicated in sperm capacitation [76,87,88,89].

On the other hand, VDAC3, despite its co-localization with VDAC2, does not seem to participate in the processes of capacitation and the acrosome reaction [71]. This is evident because spermatozoa from VDAC3 KO mice can undergo capacitation and the acrosome reaction. However, they cannot penetrate the zona pellucida, likely due to a defect in sperm motility, attributable to their shortened sheaths [71]. Table 1 shows the main information regarding the three VDAC isoforms.

## 7. Gene Expression Levels and DNA Methylation Profiles of VDAC Genes in Human Testis Cells

Some open access archives, such as the Infertility Disease DataBase–IDDB (https://mdl.shsmu.edu.cn/IDDB/ accessed on 18 June 2024) [90], FertilityOnline (https://mcg.ustc.edu.cn/bsc/spermgenes2.0/index.html accessed on 18 June 2024) [91] and Male Fertility Gene Atlas–MFGA (https://mfga.uni-muenster.de/ accessed on 18 June 2024) [92], collect and disseminate transcriptomic studies on testis cells (germ and somatic cells) and sperm for the prediction and assessment of the genetic and epigenetic association of several genes with human male fertility potential [93]. Among these, the FertilityOnline database contains information about *VDAC* expression levels in testicular single-cell RNA-seq (scRNA-seq) datasets, GSE106487 [94] and GSE120508 [95], generated by two widely used scRNA-seq platforms, Smart-seq2 and 10x Chromium, respectively. Comparing both datasets, there are similar patterns of expression profiles of *VDAC* genes analyzed in somatic cells and germs cells of the testis (Figure 1). These results are in line with the literature documenting that isoform 2 and isoform 3 could have implications in male fertility [24,25,88]; in fact, both appear to be upregulated during spermatogenesis compared to isoform 1, which is more active in somatic cells, i.e., Sertoli cells, involved in gamete development [24,25,96]. Although the *VDAC* genes are not included as candidates influencing male fertility, and neither are their possible implications discussed in these RNAseq datasets [94], it should be outlined that the independent transcriptional analyses of them in the adult human testis converge on the distinct *VDAC* gene activation process during the developmental and differentiation program of the male germline, leading to possible new mechanistic insights into the isoform-dependent regulation of spermatogenesis, sperm maturation and fertilization. The transcriptional program of *VDAC* genes in testicular cells may be epigenetically regulated by DNA methylation. To date, no studies have investigated a putative direct correlation, as *VDAC* genes are less explored research subjects in male fertility. However, only one study examining the DNA hypermethylation of the promoter of *VDAC2* associated it with idiopathic asthenospermia and low sperm motility [76]. Using a reverse analysis approach, it is possible to mine genome-wide single-cell DNA methylation profiles of *VDAC* genes in human sperm from two different datasets, GSE81233 [97] and GSE100272 [98], provided by the single-cell Methylation Bank [99] available in Methbank v.4.0 (https://ngdc.cncb.ac.cn/methbank/ accessed on 18 June 2024). According to these data, *VDAC2* and *VDAC3* result as the most hypomethylated isoforms (Figure 2), likely reflecting the trend in the transcriptional state observed in the previous datasets (Figure 1). 

## 8. Conclusions

The availability of a large set of data concerning the expression and epigenetic modifications in any kind of tissue is now confirming the regulated and selective activation of VDAC isoform genes dependent on the spermatogenesis stage. Further work is, however, necessary to establish, on a biochemical basis, the localization of the isoforms in the outer mitochondrial membrane areas of spermatozoa, where, due to the anatomical structure of the sperm, they appear to make contact with different sections, i.e., with the ODFs themselves and the very nearby plasma membrane. Moreover, the current understanding of VDACs in fertility is primarily derived from studies utilizing animal models. Future research should prioritize an exploration of the role and significance of VDACs in human fertility. The results will shed light not only on spermatozoa physiology and pathology but also on the functions of VDAC isoforms, adding to their well-known pore-forming activity.

## Figures and Tables

**Figure 1 biomolecules-14-01290-f001:**
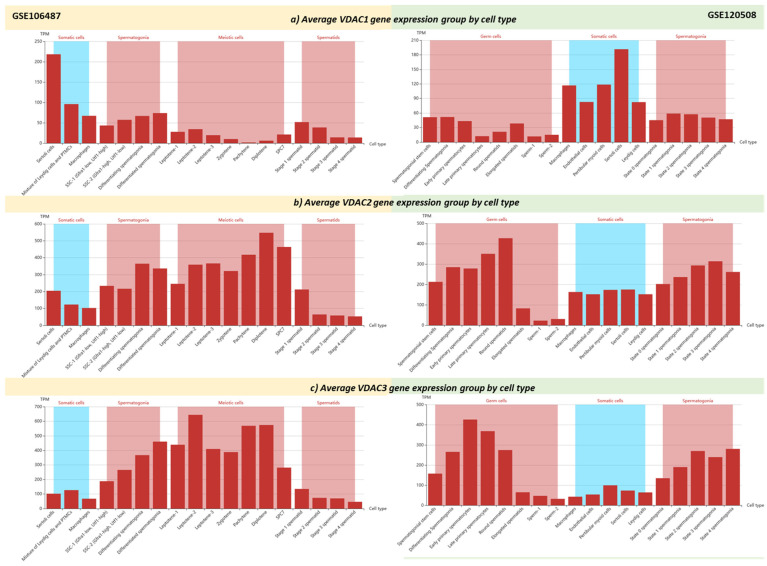
(**a**–**c**). Average gene expression profiles of *VDAC1* (**a**), *VDAC2* (**b**), and *VDAC3* (**c**) from testicular scRNA-seq data. Transcriptomic data generated by Smart-seq2 (dataset GSE106487, left) and by 10x Chromium (dataset GSE120508, right) are available from FertilityOnline database (https://mcg.ustc.edu.cn/bsc/spermgenes2.0/index.html accessed on 18 June 2024). Two types of testis cells are analyzed; somatic cells in turquoise, and germ cells in pink. TPM (transcript per million).

**Figure 2 biomolecules-14-01290-f002:**
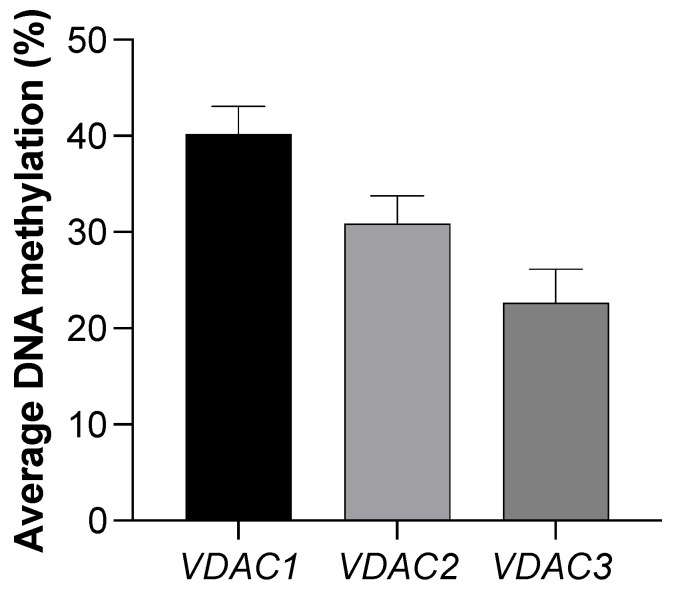
A comparison of mean distribution of average percent DNA methylation of *VDAC1*, *VDAC2*, and *VDAC3* in human sperm. Methylation data were retrieved from GSE81233 and GSE100272 datasets provided by scMethBank (https://ngdc.cncb.ac.cn/methbank/scm/ accessed on 18 June 2024). Histograms show the mean of the average DNA methylation measured in 20 samples (15 samples of GSE100272 and 5 samples of GSE81233). The values of the genes present in all samples were considered, excluding those observed in only a few samples (GSM2481616, GSM2481617, GSM2986299, GSM2986300, GSM2481655, GSM2481654, GSM2481667 of GSE81233 dataset and GSM2741043, GSM2741040, GSM2676909, GSM2676908, GSM2676907, GSM2741039, GSM2741044, GSM2741046 of GSE100272 dataset). Data are shown as mean ± SEM.

**Table 1 biomolecules-14-01290-t001:** VDAC protein isoform distribution and functions in the male reproduction system and implications in fertility.

	Knocked-Out Phenotype	Localization	Acrosome Reaction	Capacitation	Ref
VDAC1	Fertile	Sertoli cells and spermatozoa; cellular localization not clear	NO	NO	[25,68]
VDAC2	Infertile, small testis lacking mature spermatozoa	Germ cells, mature spermatozoa and testis; sperm flagella, ODFs, plasma membrane, and acrosomal region	YES	YES	[25,69,70,75,77,87,88]
VDAC3	Infertile and sperm immobility, abnormal mitochondrial sheath, spermatozoa bending, and normal spermatozoa metabolism	Germ cells, mature spermatozoa and testis; sperm flagella, ODFs, and acrosomal region	NO	NO	[25,26,71]

## Data Availability

Data supporting reported results can be found in the manuscript.

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
