# Peer review of "Voltage-Dependent Anion Channels in Male Reproductive Cells: Players in Healthy Fertility?"

_biomolecules, 2024, doi:10.3390/biom14101290_

Round 1

Reviewer 1 Report

Comments and Suggestions for Authors

The review article “VDACs in male reproductive cells: players in healthy fertility?” guides the reader from infertility and the role of mitochondria present in spermatozoa to voltage-dependent anion-selective channels in the mitochondrial membrane, of which three isoforms are related to their expression, localization, involvement spermatogenesis and its role in infertility is an interesting review that, in addition to reviewing the literature, also analyzes the expression of VDACs isoforms in databases and provides new information. The methylation status of these genes in human sperm will also be analyzed. I consider this article suitable for publication in the journal Biomolecules.

Minor concerns:

1) The sentence in line 212 only investigate about VDAC3 polymorphisms, the cite 67  is not related with the sentence. Please  review

2) Cite 66  is repeat in 77, please correct in the text in line 289

Author Response

Q1. The sentence in line 212 only investigate about VDAC3 polymorphisms, the cite 67 is not related with the sentence. Please review

R1: We agree with you with the incorrect reference. The ref. n.67 has been removed as you see highlighted in yellow in line 228, maintaining the ref. n. 66 now 76.

Q2. Cite 66 is repeat in 77, please correct in the text in line 289.

R2: Thanks for your advice. We have removed the duplicated citations and updated the bibliography (see the change highlighted in yellow in line 315).

Reviewer 2 Report

Comments and Suggestions for Authors

Stefano Conti Nibali1, et al. have review the topic named " VDACs in male reproductive cells: players in healthy fertility?". Actually, this review is crucial for some investigators to explore the mechanism of VDACs effects on the spermatogenesis or spermiogenesis and even on the sperm maturation. 

There are two points in this review need the authors to elucidate or clarified in order to improve the quality of the manuscript as follows.

1. Page222-224, “The sperm tail, indeed, can be subdivided into two compartments: the midpiece, which contains all mitochondria surrounding the ODF and the axoneme, and the principal piece, which contains the fibrous sheath, ODF and the axoneme.” This sentences description was not correct. As we know, the structure of mammalian spermatozoa can be divided into four pieces: neck, midpiece, principal piece and end piece. 

2. The authors need to present the relationships between the VDACs epression or function and Leydig cells, which are the key cells for spermatogenesis and sperm maturation, and also located in the testis and relevant to the steriodogenesis.   

Comments on the Quality of English Language

Stefano Conti Nibali1, et al. have review the topic named " VDACs in male reproductive cells: players in healthy fertility?". Actually, this review is crucial for some investigators to explore the mechanism of VDACs effects on the spermatogenesis or spermiogenesis and even on the sperm maturation. 

There are two points in this review need the authors to elucidate or clarified in order to improve the quality of the manuscript as follows.

1. Page222-224, “The sperm tail, indeed, can be subdivided into two compartments: the midpiece, which contains all mitochondria surrounding the ODF and the axoneme, and the principal piece, which contains the fibrous sheath, ODF and the axoneme.” This sentences description was not correct. As we know, the structure of mammalian spermatozoa can be divided into four pieces: neck, midpiece, principal piece and end piece. 

2. The authors need to present the relationships between the VDACs epression or function and Leydig cells, which are the key cells for spermatogenesis and sperm maturation, and also located in the testis and relevant to the steriodogenesis.   

Author Response

Q1. Page222-224, “The sperm tail, indeed, can be subdivided into two compartments: the midpiece, which contains all mitochondria surrounding the ODF and the axoneme, and the principal piece, which contains the fibrous sheath, ODF and the axoneme.” This sentences description was not correct. As we know, the structure of mammalian spermatozoa can be divided into four pieces: neck, midpiece, principal piece and end piece.

R1. Thanks for your rectification. We have changed the sentence as you suggested in line 238-239, highlighted in yellow.

Q2. The authors need to present the relationships between the VDACs expression or function and Leydig cells, which are the key cells for spermatogenesis and sperm maturation, and also located in the testis and relevant to the steroidogenesis.  

R2.  According to your recommendation, we have added information about VDAC in Leydig cells (Lines 246-254). This new data may expand a little bit the relationship between the VDACs expression or function and Leydig cells, since in the previous version of our Ms. we have limited to mention the only validated evidence of VDAC3, as reported in line 229.

Reviewer 3 Report

Comments and Suggestions for Authors

In paragraph 2 it should be mentioned that sperm mitochondria contain LDHx which is not exist in the mitochondria of somatic cells. Also it was published that sperm ribosomal mitochondria can translate proteins by nuclear encoded mRNA. 

Author Response

Q1. In paragraph 2 it should be mentioned that sperm mitochondria contain LDHx which is not exist in the mitochondria of somatic cells. Also it was published that sperm ribosomal mitochondria can translate proteins by nuclear encoded mRNA.

R1. Thanks for your precious suggestions. We have included information about LDHs (lines 94-103); moreover, we have mentioned the phenomenon of the nuclear-encoded proteins by mitochondrial-type ribosomes (Lines 114-120).